# General policy mapping: online continual reinforcement learning inspired on the insect brain

**Angel Yanguas-Gil**
Applied Materials Division
Argonne National Laboratory
Lemont, IL 60062
ayg@anl.gov

**Sandeep Madireddy**
Mathematics and Computational
Science Division
Argonne National Laboratory
smadireddy@anl.gov

## Abstract

We have developed a model for online continual or lifelong reinforcement learning (RL) inspired on the insect brain. Our model leverages the offline training of a feature extraction and a common general policy layer to enable the convergence of RL algorithms in online settings. Sharing a common policy layer across tasks leads to positive backward transfer, where the agent continuously improved in older tasks sharing the same underlying general policy. Biologically inspired restrictions to the agent's network are key for the convergence of RL algorithms. This provides a pathway towards efficient online RL in resource-constrained scenarios.

## 1   Introduction

Insects exhibit a remarkable ability to learn new tasks. In addition to their well-documented abilities in the wild, researchers have taught insects such as honeybees and bumblebees a wide range of tasks, from learning to manipulate objects to access food, to navigating a maze using visual cues, to "playing soccer".(1; 11) These capabilities are particularly relevant in the context of online reinforcement learning (RL) and, in particular, in situations where a single agent must learn from the interaction with its environment in as few examples as possible. Convergence to optimal policies under these assumptions is extremely challenging within the framework of Markov decision processes (MDP).(14) Therefore, the question is: How can insects accomplish this?

In this work, we have formulated a model, which we refer to as *general policy mapping*, that is consistent with the current understanding of insect neuroscience. At the core of this approach is an effective partitioning of the problem into offline and online phases that greatly simplifies the online learning of new tasks. In particular, the policy network is partitioned into three different components, each of which has a well-defined counterpart in the insect brain: a feature extraction layer, an online policy mapping layer, and a general policy layer shared across tasks.

We have evaluated this algorithm against a series of 1 in 5 target discrimination tasks implemented in a custom open world first person point of view environment. In these tasks, the agent is shown 5 objects and needs to learn to reach the correct target. Through sampling over multiple seeds, we have demonstrated the convergence of a simple cross entropy algorithm when the agent is only allowed to interact with a single environment under sparse reward conditions. The same algorithm fail to converge when presented with the general task in the absence of offline contributions. Sharing a common general policy across tasks also lead to positive backward transfer in continual RL scenarios.

Offline Reinforcement Learning Workshop at Neural Information Processing Systems, 2022.

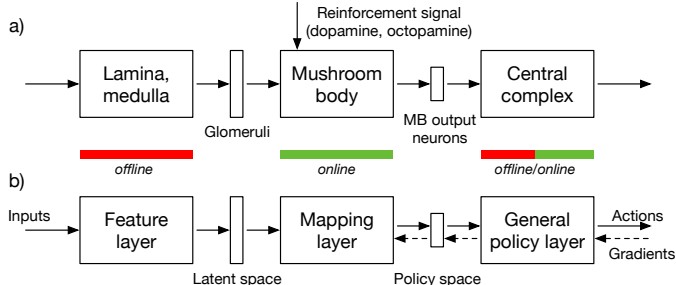

Figure 1: a) High level diagram of online learning pathways in hymenopterans (i.e. bees and ants) b) General policy mapping model explored in this work

## 2  Model

In hymenopterans, such as bees and ants, visual inputs are processed in a way that is reminiscent of the visual processing system in mammals. This internal representation, which is encoded in the optical glomeruli, is then passed into the mushroom body, which is the learning center of the insect brain (Fig. 1). The mushroom body (MB) plays a critical role in online learning, with ablation experiments showing that the absence of a functional MB prevents insects from learning new tasks while preserving their ability to carry out innate tasks.(2; 7) The MB aggressively reduces the amount of data being processed, transforming large-dimensional inputs into a low-dimensional space. Moreover, despite its central importance, the number of connections coming out of the MB is relatively small. The consensus is that the MB does not encode policies, but instead encodes saliency and valence in a task-dependent fashion. This information is then passed to other centers in the insect brain, where it is used to drive insect behavior.(5)

From an RL perspective, this structure is consistent with a model in which the input space $\mathbf{s}$ is subsequently transformed into a latent space $\mathbf{s}_l$ and a policy space $\mathbf{s}_p$, which dictates which actions the insect is taking based on a set of policy drivers:

$$\mathbf{s} \to \mathbf{s}_l \to \mathbf{s}_p \to a \tag{1}$$

This structure also strongly resembles a partially observable Markov decision process,(9) with the feature and mapping layers effectively transforming observations into an internal representation space that is used to formulate a more general policy.

The general policy mapping approach introduced in this work implements this type of structure, splitting the policy network into three different components (Fig. 1): a *feature layer*, a *mapping layer*, and a *general policy layer*. Both the feature and general policy layers are pretrained offline, with the mapping layer being responsible for mapping online tasks to the policy layer.

In order to strengthen the roles of these three components, we have introduced two key restrictions that are present in the insect brain: 1) the mapping layer behaves as a saliency layer, with the output being a linear combination of multiple channels in which both inputs and weights are restricted to be positive. This is consistent with the preservation of the type of synapse (in this case excitatory) during the online learning process. As we will show later, this restriction turns out to be critical for the convergence of single environment RL algorithms. 2) The policy layer is restricted so that, in absence of inputs, the output reverts to a default policy, which in our case we chose it to be a random policy. Again, this ensures that the outputs from the mapping layer act as drivers that shift the policy towards specific goals and encourages exploration when facing a new task.

Finally, while the mapping layer is task-specific, the general policy layer is shared among all tasks.[Fig. 2(a)] Enabling plasticity in the general policy layer during the online phase lets the agent refine its general policy as it faces new online tasks. This is markedly different from the multiple head approach used in task-incremental continual learning, where only features are shared across tasks. This feature is critical to achieve positive backward transfer in continual RL scenarios.

# 3 Background

## 3.1 Continual reinforcement learning

Continual RL involves a stream of tasks where each task can be considered a stationary Markov decision process (see Ref. (10) for a detailed survey of this field). There are two complementary views of continual RL: the non-stationary function view considers each task as an independent, stationary MDP $M^{(z)}$ characterized by its own independent parameters. The partially observed view considers each task as a new, potentially unobserved component of a larger, stationary, partially observable Markov decision process.(17) This second view is consistent with the experience of an insect over its lifetime: if we take foraging as an example, each time that an insect encounters a new type of food has to first identify the new object as an edible resource, and optimize the way it manipulates it.

Ref (10) provides a useful taxonomy of continual RL approaches, which at its highest level can be broken down into three categories: approaches exploiting explicit knowledge retention, those that leverage shared structure, and those focused on learning to learn. This work can be framed in the context of the first two categories: the use of task-specific mapping layers is a common approach to retaining knowledge when the agent is task-aware. On the other hand, the use of a common general policy layer in our work aims at reusing a common underlying structure. This resembles approaches that try to enforce a common representation across tasks (i.e. Ref. (3)).

## 3.2 Transfer and meta reinforcement learning

Millions of years of evolution have shaped the architecture and learning ability of insects. These can be viewed as a massive offline component that has optimized insects to excel at online reinforcement learning tasks. Transfer and meta-reinforcement learning provide the appropriate context when looking at the role that insect architecture can have in jumpstarting their online and continual learning abilities.(15) A key difference between transfer RL approaches and this work is that, due to the initialization of the policy mapping layer at the beginning of each task, we do not expect to see a significant jumpstart as the agent encounters new tasks, which is one of the traditional metrics used to quantify transfer learning.(15) We do expect to see an acceleration of the learning process due to the offline learning component.

In traditional metalearning assays, an agent is trained to learn tasks extracted from a given distribution of tasks during a meta-training phase, and then evaluated on its ability to learn new, previously unseen tasks from a newer distributions. There is clear parallelism between this approach and the offline training component explored in this work. Algorithms such as SNAIL have shown to greatly accelerate learning of new reinforcement learning tasks.(12) However, a key difference is that we focus on online lifelong learning scenarios where the agent is expected to see a stream of tasks and where parts of the agent continue to evolve over its lifetime.

Finally, a fundamental difference between Markov decision processes and biological systems is that the discount factor $\gamma$ does not appear explicitly in the interaction of an individual with their environment: instead, works emphasizing the connection between MDPs and neuroscience treat the discount factor as a parameter that is influenced by variables internal to the agent, such as the concentration of different neuromodulators.(8; 14) One potential interpretation is that the evolutionary process has lead to a neurochemistry that is optimized to learn the type of tasks relevant for an individual in the wild. This results on agents that are proficient at learning stream of tasks without the need of an explicit discount factor. This is an issue still largely unexplored in the context of lifelong reinforcement learning, with the majority of examples of meta-RL algorithms in the literature still operating under a full MDP assumption where the discount factor is externally provided for each task (i.e. Refs. (12; 4)) In this work we sidestep this issue by focusing on algorithms, such as the cross-entropy approach, that minimize the need of prior information about the task at hand.

# 4 Experiments

The core hypotheses we want to explore are whether the policy mapping approach enables simple RL algorithms to reliably converge to optimal policies under online RL restrictions, and whether it promotes continual/lifelong learning through the sharing of a common general policy across tasks.

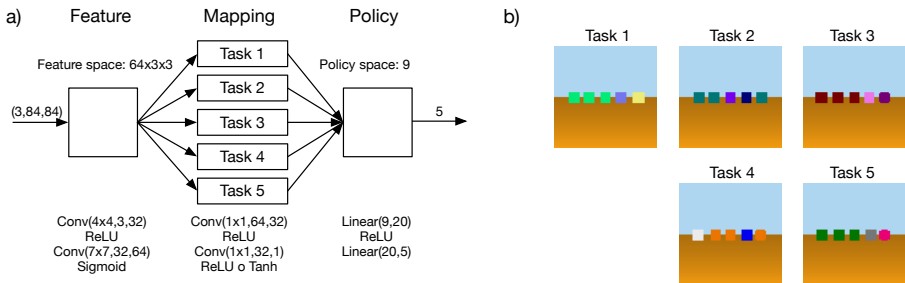

Figure 2: a) Policy matching network structure b) Snapshots of 1 in 5 object identification task implemented in our custom environment RLBug

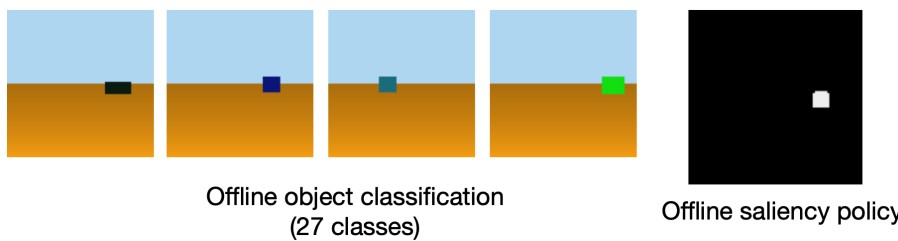

Offline object classification
(27 classes)

Offline saliency policy

Figure 3: Samples for the offline classification and offline reinforcement learning tasks. The feature extraction layer is trained on a classification task of randomly placed objects, while the general policy layer is trained on a task comprising reaching an object based on a saliency map.

To this end we designed a series of 1 in 5 object identification tasks where the agent has to reach a task-specific target, and implemented in RLBug, a custom first person point of view, open world environment developed specifically for this work. Snapshots of five different tasks are shown in Figure 2(b). Inputs are 3 channel 84x84 images, and the agent's action space comprises 5 possible actions (moving forward, stationary turns, and turning while moving). In order to mimic the type of scenarios that insects would find in the wild, a constant positive reward is offered only if the agent reaches the right object before the end of the episode. The positions and ordering of the different objects are randomly selected for each episode to ensure that the agent learns to identify the right target object rather than going to a specific location. Episodes only terminate prematurely if the agent reaches any of the objects. Otherwise, the agent is allowed to wander freely until reaching a maximum number of steps. This further adds to the difficulty of the task. Details of the environment can be found in the Appendix.

The experiments comprise an offline and an online training phase: the offline phase involves the training of both the feature and the general policy layers. The feature layer is trained using an object classification task comprising 27 different objects of random sizes placed in random positions in front of the agent (see Figure 3 and the Appendix for details). The resulting feature extraction layer is transferred directly to the RL agent. The general policy layer is trained using a baseline task where the agent needs to reach a single object, using a 84x84 binary saliency map as input. The head of the trained network is then used as the general policy layer for the online learning phase.

The online phase of the experiment involves training the agent in 5 1 in 5 object identification tasks. We have trained the agent using the cross-entropy algorithm(6) with an additional entropy term.(13; 16) This algorithm is known to struggle when tasks are too complex, but it requires the least amount of prior information on the tasks, which is consistent with online scenarios where tasks are not known in advance. Consequently, the convergence probability is one of the key metrics that we have explored in this work, using 20 randomly selected seeds per task to generate a distribution of learning curves. For the continual learning setting, we trained the agent consecutively on the five tasks in a task incremental setting.

Table 1: Convergence of cross entropy algorithm for the full general policy matching model as well as selected ablated configurations for the 1 in 5 object selection task. Each data point is the average of 100 independent runs across five different tasks shown in Figure 2(b)

| | Cross entropy algorithm | |
| --- | --- | --- |
| Configuration | Convergence | # Episodes, av(std) |
| Fixed general policy | 100% | 530 (240) |
| **Adaptive general policy** | **99%** | **280 (80)** |
| No offline policy | 94% | 820 (320) |
| No offline feature | 30% | 740 (340) |
| No offline components | 10% | 1200 (420) |
| Unclamped weights | 70% | 230(130) |

## 5  Results

### 5.1  Offline reinforcement learning

Training the general policy layer on the offline saliency task is a trivial process, resulting on an optimal policy $\pi(\mathbf{s}_p)$ connecting the policy space with the action space of the agent. Convergence was observed in nearly all cases for the cross-entropy, reinforce, and actor-critic algorithms, taking an average of 200 episodes to converge. Likewise, training of the feature extraction layer on the classification task resulted on a 98% accuracy after 20 epochs. The trained feature extraction and general policy layers were then used to jumpstart the online reinforcement learning.

### 5.2  Online reinforcement learning

The results obtained show that the general policy matching model enables a simple RL algorithm such as cross entropy to converge to the optimal policy with high probability. (Table 1). The values shown in the Table have been obtained after averaging a total of 100 runs (20 per each of the tasks). Ablation studies show that the offline training of the feature layer and the clamping of the weights in the mapping layer are key to achieve high levels of convergence.

Offline RL training to a general policy using a surrogate task significantly decreases the number of episodes required for learning with respect to the case in which the policy layer is not pretrained, reducing the average number of episodes required at least by a factor of three. As a reference, a deep Q learning algorithm trained on a similar network requires an average of 1200 episodes.

Finally, we also examined the case with no offline training in ether the feature or the general policy component. In this scenario, the convergence achieved with the cross entropy algorithm is really poor.

### 5.3  Online continual learning

We have then taken the two top performing configurations, the policy matching model with and without online adaptive policy, and applied them to a task-incremental continual learning task where a single agent is exposed to the five 1-in-5 tasks consecutively. In between each task we evaluated the agent's performance on each of the tasks to track the impact that learning newer tasks had on the agent's ability to learn older ones. We used the same cross-entropy algorithm to train the agent in each of the tasks without any additional strategies involving either synaptic consolidation or replay to preserve information of prior tasks.

In Figure 4 we show the evolution of the agent's accuracy averaged over the last twenty episodes over as it is trained on each of the five tasks. The transition between tasks are indicated by the vertical lines. Changing to a new tasks naturally leads to a drop in accuracy, as expected by the fact that at the beginning of each task the agent uses an untrained policy mapping layer.

In order to better understand the evolution of the agent's abilities we have evaluate the agent's performance in all the tasks after being trained in each of the tasks. The results, shown in Figure 5, show that even when we share the general policy layers among the tasks, the agent is capable

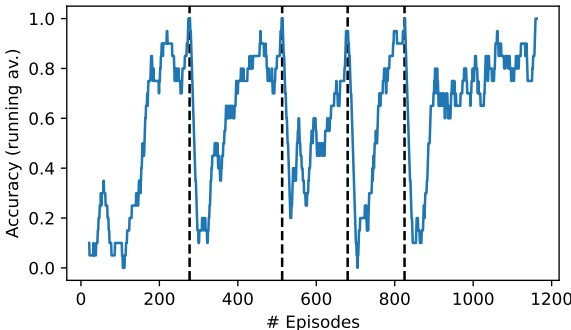

Figure 4: Evolution of the agent task accuracy over a single run where the agent is consecutively trained in five different tasks. Vertical lines mark the transition between tasks.

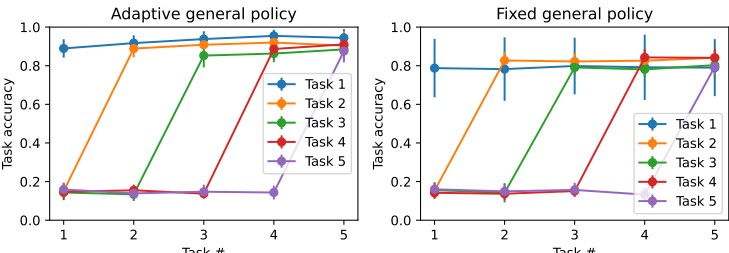

Figure 5: Task accuracy in a task-incremental continual RL test involving Tasks 1 to 5. Results are shown for two agents: policy matching agent with an adaptive general policy and with a fixed (offline) general policy. Results shown are the average of 20 separate runs

of sequentially learning multiple tasks without any evidence of catastrophic forgetting. In fact, the performance on earlier tasks seem to increase as it is trained on newer tasks [Figure 5(a)], which is an evidence of positive backwards transfer. In absence of online learning in the general policy layer, the agent's performance in older tasks remains stationary as it is trained on newer tasks [Figure 5(b)]. The performance using a baseline deep-Q learning algorithm show a significant amount of catastrophic forgetting, as shown in the Appendix. In this case, a buffer was restricted to each individual task to eliminate the agent's ability to train on data from older tasks.

As in the single task case, the agent with an adaptive general policy requires a lower number of episodes to learn each of the tasks (Table 2). We do not observe any significant decrease in the number of episodes required to learn later tasks in the sequence, which is consistent with learning in the mapping layer being the bottleneck for the agent. This is expected given that 1) the offline training on the saliency task helps jumpstart the training on the online tasks, as shown on the ablation studies in Table 1; and 2) the mapping layer is initialized at the beginning of each new task and there is no overlap between the objects involved in each of the tasks.

Table 2: Average number of episodes required to learn each of the five tasks during the online continual RL test for the general policy matching agent with an adaptive general policy and a fixed (offline) general policy.

| | # of Episodes, average (std) | | | | |
| --- | --- | --- | --- | --- | --- |
| | Task 1 | Task 2 | Task 3 | Task 4 | Task 5 |
| Adaptive general policy | 250 (50) | 190 (50) | 252 (80) | 280 (140) | 200 (70) |
| Fixed general policy | 500 (300) | 400 (100) | 600 (200) | 480 (140) | 720 (180) |

# 6 Conclusions

In this work we have explored a model for online continual RL inspired on the insect brain. The results obtained show that this type of architecture promotes convergence of simpler algorithms that are well suited for online RL scenarios where no prior information of the specific tasks is available. The best performing configurations also excel in continual learning tasks, with the adaptive general policy case actually improving its performance on older tasks as the agent is able to refine its underlying general policy. Here we have focused on a set of tasks that resemble the type of learning that insects would be required to do as it explores an open world environment. An implementation of roundworld and the RL tasks used in this work can be found at `https://github.com/anglyan/roundworld`.

A fundamental challenge of online lifelong RL is that, in the real world, tasks do not often come with an externally provided discount factor. In this work we have sidestepped this problem by focusing on the cross-entropy method. However, generalizations of this approach need to consider how to leverage offline meta RL approaches to identify optimal values of the discount factor for a given distribution of tasks.

Finally, in addition to shedding some light on how biological constraints can promote online RL, we believe that the model explored in this work can be valuable to explore novel hardware architectures for online, on chip lifelong learning under resource constrained scenarios. In particular, heterogeneous plasticity and the absence of any replay approaches are two key features that are compatible with lightweight hardware implementations.

## Acknowledgments and Disclosure of Funding

This research was funded through DARPA's Lifelong Learning Machines program.

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

# A  Appendix

## A.1  RLBug environment

RLBug is a RL environment that implements the interaction of an agent from a first person point of view perspective in an open-world environment. RLBug is implemented as a pure Python environment and it is extremely lightweight, capable to achieve more than 500 frames per second in a single core for an scenario comprising 20 objects. Our goal was to find a balance between the need to develop a visually rich set of tasks while avoiding the slower frame rates of near real time environments based on sophisticated engines, which are harder to implement in high performance computing settings.

Tasks are generated by passing a configuration of objects in which each type of objects is defined by color, height, width, and the reward of the object. The color of the ground, sky, objects, and other effects such as depth-fading, are all customizable, as is the type of output (either 1 channel or 3 channel images). For this work, we considered a state comprising an RGB 3-channel 84x84 pixel image encoded as a (3,84,84) array, and an action space with five actions: moving forward, turning left and right, and turning while moving. RLBug will be made publicly available by the time of the workshop.

In this work we used RLBug to implement a 1 in 5 object identification task. In this task the agent is presented five objects, one target object, one decoy object, and three instances of background objects. The agent must learn to move and reach the target object within a maximum number of steps. In order to prevent the agent from learning a deterministic policy, the objects are placed at distance from the agent that is randomly determined for each episode around a mean value. The ordering of the objects as seen by the agent are also randomized for each episode. A positive reward is provided only when the agent reaches the target within the predetermined number of steps.

A key difficulty of this task is that the agent is free to roam in the 2D environment until the maximum number of steps is reached. This means that, if the agent loses track of the objects, the agent is seeing an empty landscape as input. Baseline studies using deep Q learning and various policy gradient methods show that this reduces the algorithms ability to converge with respect to the case where the episode ends once none of the 5 objects is within the agent's visual field. However, we feel that this type of open world interaction reproduces the type of interactions that insect have with their environment and is also representative of many potential applications where an agent has to learn to carry out tasks in the world without any external cues.

## A.2  Offline training of the feature layer

Insects do not learn everything from scratch, but instead they come with a well-defined architecture for processing visual inputs. In insect this is carried out primarily in the lamina and medulla, which have evolved to maximize the insect's ability to learn tasks that are not known beforehand across multiple domains.

The problem of how to identify an optimal set of features that maximizes an agent's ability to quickly learn new RL tasks after deployment is currently one of our active areas of research. For this work, though, we focused on a single architecture that is inspired on some of the design principles in the lamina and medulla of insects that we have trained on a single downstream task. A key characteristic is that the output layer comprises several non-overlapping large field areas with many channels each.

For this work, we considered a feature extraction network comprising two convolutions, one 4x4 layer producing 32 channels, and one 7x7 layer outputting 64 channels. A rectified linear unit and a sigmoid function were used for the first and second convolutional layers (Figure 2). In order to train the feature extraction layer, we added an all-to-all linear layer as output layer. We trained this network in an object classification task comprising 27 different types of objects. We generated a

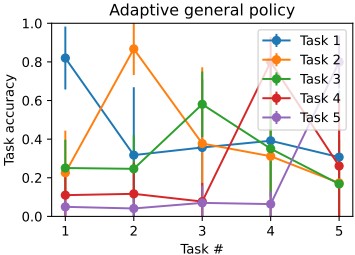

Figure 6: Task accuracy during the continual learning of Tasks 1-5 using deep Q learning. The results are consistent with a significant amount of catastrophic forgetting.

dataset of images in which we placed an object of a random size and width on a random location within the agent's visual field. We then partitioned the RGB space into equally sized regions and generated objects using randomly selected colors within each of these regions. The resulting dataset comprises 500 images per category with 100 additional images reserved for testing. We trained the network against the training dataset, achieving 98% accuracy in the testing dataset after 20 epochs. The feature extraction component was then used in the RL tasks.

### A.3  Deep Q learning baseline results

As a baseline we used a deep Q learning algorithm to sequentially train the agent on Tasks 1 to 5. In order to prevent cross-talks between the tasks we started each Task with an empty buffer. For this baseline case we used the pretrained feature layer and we maintained the same matching and general policy layers used for the cross-entropy method. The key difference is that now the network is used to learn the state-action values $Q(s, a)$, and we applied an $\varepsilon$-greedy approach to compute our policy. The results, shown in Figure 6, are consistent with the presence of catastrophic forgetting, with the agent increasingly forgetting the learned tasks. A cut-off number of 2000 episodes is used for each tasks, with the learning process ending earlier if the agent's performance exceeds 95% in the last 25 episodes under a value of $\varepsilon = 0.02$. The higher error bars are due to the variance in the learning process.

