# OpenReview forum: "General policy mapping: online continual reinforcement learning inspired on the insect brain"
_NeurIPS.cc/2022/Workshop/Offline_RL — Offline RL Workshop NeurIPS 2022_

### Official Review · Reviewer_93Ez · 2022-10-13
**Neat ideas; unclear how offline RL is actually used**

**Rating:** 7
**Confidence:** 5

**Review:**

I found this paper to be a neat introduction to some interdisciplinary concepts that embody this year's workshop theme "Offline RL as a launching point" really well. I just wish that the offline RL component of pretraining the general policy layer was more detailed.

The demonstration of positive backward transfer when this pre-trained policy layer is adapted after each task is a great result. It was also nice to see that if one was unable to adapt the policy layer that performance on prior tasks doesn't degrade as new tasks are observed. Perhaps the greatest contributor is through the training of the mapping layer for each task? This is a good argument for the relative strength of the offline RL pre-training component. I am however disappointed that the DQN was not fairly trained as an adequate comparison (where the replay buffer was unable to access information from prior tasks). Was the DQN pre-trained with offline RL at all or was it primarily used as a completely online RL comparison?

There are two additional recommendations that I have:
- It would be great to see the learning curves of each task underlying Figure 3. I'm curious whether there are added efficiencies with each successive task being able to use the general policy layer that is continually improved. What I mean is, does each individual task converge in fewer episodes with each task observed?
- The grounding of this paper is not sufficiently based in relevant model-based/continual/latent-variable methods for transfer or continual learning in RL. It would be great if more of these papers were used as conceptual foundation of this approach. A small collection of such works here that the authors may find informative:
  - "Direct Policy Transfer"; Jiayu Yao, et al (2018)
  - "Efficient Off-Policy Meta-Reinforcement Learning via Probabilistic Context Variables"; Kate Rakelly, et al (2020)
  - "Fast adaptation via policy-dynamics value functions"; Roberta Raileanu, et al (2020)
  - "Generalized hidden parameter mdps: Transferable model-based RL in a handful of trials"; Cristian Perez, et al (2020)

---

### Official Review · Reviewer_uvLn · 2022-10-20
**A good paper with interesting motivation, but with limited experimental results**

**Rating:** 6
**Confidence:** 3

**Review:**

This paper proposes a framework to first learn a feature representation, and then deploy the learned representation to solve multi-task RL problems. This paper also compares such intuition of learning representation for multi-task with biology. The reviewer thinks such a connection between RL and biology is interesting and intuitive, but the experimental results are a bit limited (only on toy environments). In the future, it would be exciting to see how the proposed method can be scaled up to larger datasets and harder tasks!